# Impact of Injury Frequency and Severity on Mental Health Indicators in Triathletes: A Repeated-Measures Study

**DOI:** 10.3390/healthcare13141657

**Published:** 2025-07-09

**Authors:** Laura Gil-Caselles, Roberto Ruiz-Barquín, José María Gimenez-Egido, Alejo Garcia-Naveira, Aurelio Olmedilla-Zafra

**Affiliations:** 1HUMSE Research Group, Faculty of Sport Sciences, University of Murcia, 30100 Murcia, Spain; laura.gilc@um.es (L.G.-C.); josemaria.gimenez@um.es (J.M.G.-E.); 2Department of Education, University of Autónoma Madrid, 28049 Madrid, Spain; roberto.ruiz@uam.es; 3Faculty of Psychology, University of Villanueva, 28034 Madrid, Spain; 4Humse Research Group, Faculty of Psychology, University of Murcia, 30100 Murcia, Spain; olmedilla@um.es

**Keywords:** triathlon, injury, anxiety, depression, stress, mood state

## Abstract

Background: The complexity of triathlon goes beyond the multidisciplinary nature of the sport and extends to the physical and mental health of the athlete. One of the most relevant aspects is injuries, which, in addition to the physical impact, can affect mental health indicators. Objective: The objective of this study was to determine the relationship between injuries sustained by triathletes and mental health indicators. Methods: Sixty-three subjects participated, of whom 48 suffered one or two injuries. The average age was 37.83 years, and the sample consisted of 34 men (39.56 years) and 29 women (32.21 years). The instruments used were an online questionnaire to collect the number of injuries, type, and severity; the Depression, Anxiety and Stress Scale-21 (DASS-21); and the Profile of Mood States (POMS). A longitudinal study was conducted, lasting six months, where the questionnaires were administered monthly. Results: The greater the number of injuries, the higher the scores in the coefficients of variation of anger and vigor, and the highest peak is found in the variable stress, followed by anxiety and depression. Conclusions: Triathletes who suffer a greater number of injuries have higher scores in stress and depression, and their level of vigor and anger is increased, so they present a more negative and reactive stress and mood profile.

## 1. Introduction

In recent years, the scientific literature has increasingly addressed the relationship between sport and mental health disorders [1,2]. This study focuses specifically on triathletes, a group exposed to unique physical and psychological stressors due to the demands of training and competing in three endurance disciplines. Mental health is defined as a state of well-being in which individuals realize their abilities, can cope with everyday stress, work productively, and contribute to their communities [3]. It is a foundational element not only for personal well-being but also for the effective functioning of society. Mental health issues can affect anyone, including athletes. However, determining accurate prevalence rates remains challenging due to the predominant focus on elite populations and methodological limitations. Recent reviews estimate that between 19% and 34% of elite and semi-elite athletes may experience symptoms of common mental disorders, such as anxiety or depression [4]. Nevertheless, evidence remains scarce regarding endurance athletes, particularly triathletes, despite their exposure to prolonged physical and psychological stressors. Moreover, methodological challenges—including heterogeneous measurement tools, inconsistent cut-off points, and underreporting due to stigma—further complicate prevalence estimates.

In the context of sport, it is essential to consider the specific demands of the discipline practiced and the age at which athletes begin their involvement, as these factors influence psychological development throughout the athlete’s career. The relationship between mental health and physical activity is also bidirectional: individuals with poor mental health are less likely to engage in physical activity, thereby worsening their condition, and vice versa [5,6].

Athletes face a variety of stressors that can compromise their mental health, ranging from life stress to sport-specific challenges such as performance pressure, failure, injury, and retirement from sport [4]. In turn, mental health disorders can increase injury risk and negatively affect recovery outcomes. If left unaddressed, psychological problems can deteriorate, resulting in poorer performance and greater vulnerability to injury [7].

To understand the relationship between mental health and injuries, it is crucial to consider the sport’s modality and the associated training load. Each sport has distinct characteristics that result in specific injury patterns [8,9]. Accordingly, as Meeuwisse et al. [10] assert, the etiology of sports injuries is multifactorial, encompassing sporting, personal, physical, and psychological factors.

The bidirectional relationship between injury and mental health becomes particularly salient during rehabilitation, a critical period during which psychological aspects take on special relevance [5]. Injuries and premature dropout from sport are often linked to stress and can trigger or exacerbate mental disorders [11,12].

Triathlon, a sport combining swimming, cycling, and running [13], has seen rapid growth in recent years [14]. This increased participation has spurred research aimed at improving training and competition responses, with particular focus on psychological factors [15,16,17,18,19]. Triathletes, in comparison to other endurance athletes, are at higher risk of injuries due to overuse, overstretching, overtraining, and illness [20]. Some studies suggest that the pursuit of high training goals in triathlon can lead to negative health consequences and a high prevalence of overuse injuries [21,22], likely due to the elevated stress of simultaneously training in three demanding disciplines. When training efforts are better distributed across disciplines, physical performance tends to improve, and injury rates decline [23].

Despite their high level of physical conditioning, athletes are not immune to common mental disorders such as depression, anxiety, or stress. This makes it necessary to investigate their prevalence, particularly in demanding sports like triathlon and in vulnerable situations such as injury [5]. Stress, for example, can lead to increased muscle tension and impaired performance, while injury itself can induce elevated stress and emotional disturbances that hinder recovery [24,25,26]. These disturbances often manifest in key mental health indicators such as anxiety, depression, and mood disorders [27,28].

Mood, in particular, appears to influence both the likelihood of injury and the rehabilitation process [29,30,31]. Understanding the mood states of injured athletes is therefore vital, as it can inform their readiness for return to play and their ability to adhere to rehabilitation protocols.

Recent studies have increasingly highlighted the emotional and psychological impact of injury, especially during recovery. For instance, Ríos-Garit et al. [32] found that injured youth athletes commonly experience heightened anxiety and negative mood states during rehabilitation, emphasizing the need to integrate psychological assessment and support into recovery programs. In this vein, Olmedilla and García-Mas [33] introduced the Psycholight protocol, which uses psychological evaluation and intervention as tools for both prevention and recovery in injury management. Structured interventions such as this help identify psychological risk factors early, enhance adherence to rehabilitation, and prevent relapse.

Furthermore, Olmedilla et al. [34] argue that stress alone does not fully explain psychological vulnerability in injured athletes. Through a Bayesian framework, they demonstrated how self-confidence, resilience, and cognitive appraisal also contribute significantly to both injury susceptibility and the challenges of recovery. This supports a multidimensional approach to understanding athlete mental health.

For sport psychology professionals, it is essential to understand the psychological profile of athletes both in competitive performance and during rehabilitation. These profiles provide insights into mental skills related to performance and can inform individualized strategies for psychological support and injury prevention [35,36,37]. In the case of triathletes, in-depth study of their psychological traits is especially important due to the solitary and demanding nature of the sport [14] and the key role of psychological variables in sports performance [38,39].

Accordingly, the aim of the present study is to determine the relationship between injuries sustained by triathletes and their mental health indicators, with specific focus on how the number of injuries affects psychological well-being.

## 2. Materials and Methods

### 2.1. Participants

The participants were selected by non-probabilistic sampling based on accessibility criteria. There were 63 subjects, of whom 15 did not report any injury but still completed the online questionnaires. The remaining participants (*n* = 48) sustained at least one injury during the months of evaluation considered. The average age of the 63 athletes was 37.83 years *(SD* = 13.91). There was a balanced gender distribution, with 53.57% men (*n* = 34) and 46.43% women (*n* = 29).

Inclusion criteria for the study included the following: age category—only athletes 18 years and older, i.e., junior (18–23), senior (24–39), veteran (40–59), master (40–59), and paratriathlon; and performance level—participation in competitions at one or more of the following levels: regional, national, or international. Only triathletes who had competed in the last 12 months or during the study period were included. Exclusion criteria comprised non-participation in a triathlon or not having competed within the year prior to the study.

Although the total sample comprised 63 triathletes, the number of responses varied across the six months of data collection due to participant availability and adherence. As a result, the total number of observations across time points amounted to 121 valid data entries. This variability was accounted for in the analysis and discussed as a limitation in the corresponding section. Although the sample was not probabilistic, it is considered adequate for an exploratory study of this nature. No formal sample size calculation was conducted prior to data collection due to the accessibility-based recruitment strategy and the repeated-measures design. However, the total of 63 participants, with 121 valid entries across six time points, is in line with other exploratory studies in the field of sports psychology that employ similar sample sizes and research approaches.

### 2.2. Instruments

The assessment instruments used for the study were as follows:
Personal and sports variables questionnaire. An ad hoc questionnaire to collect socio-demographic data from the athlete. It includes personal information, club affiliation, competition category, years at the top level, number of years practicing sport, training days, and training hours (see Appendix A).History of sports injuries. An ad hoc questionnaire was developed based on an injury protocol [40]. It collects the number of sports injuries suffered in the last month and specific data on them. The variable Weighted Injury Severity Index is transformed into the TGSP (overall time margin in which the athlete is out of training or competitions due to injury throughout the entire assessment period), and the days during which the athlete is injured are added based on their severity:
Mild. When treatment is required and at least 1 day without training. Average of 2.5 days of absence.Moderate. When treatment is required for 6 days or more without training and with loss of some competition. Average of 18 days of absence.Serious. When it requires one to three months of sports leave, sometimes hospitalization, and even surgery. Average of 60 days of absence.Very Serious. When it requires more than 4 months of sports leave, it sometimes causes a permanent decrease in sports performance and constant rehabilitation. Average of 135 days of absence.

Example: An athlete who has had two moderate injuries in February: 18 + 18 = 36 days of absence; an athlete who has had 5 minor injuries in the last 6 months: 2.5 + 2.5 + 2.5 + 2.5 + 2.5 = 10 days of absence.

The categorization of injury severity and the corresponding average number of days lost were based on the methodology described in a previous study [40]. Although this weighted injury index (TGSP) has not been widely applied in triathlon-specific populations, its structure has been validated and used in other performance sports contexts and is applied here for the first time in this type of multi-discipline endurance sport. Its adaptation to triathlon offers a more accurate estimation of injury burden over time.

To assess mental health indicators, the adapted and validated Spanish version of the Depression, Anxiety and Stress Scale-21 items (DASS-21, [41]) was used. The internal consistency of the DASS-21 was a Cronbach’s α of 0.91 for the total scale and 0.85 for the Depression scale, 0.83 for the Stress scale, and 0.73 for the Anxiety scale [42]. This scale has three subscales: Depression, Anxiety, and Stress, each composed of 7 items, for a total of 21. Responses are given on a four-point Likert-type scale. In the present study, the reliability coefficients obtained were similar, with an α of 0.90 for the total scale, an α of 0.84 for the Depression scale, an α of 0.82 for the Stress scale, and an α of 0.74 for the Anxiety scale, indicating adequate internal consistency in our sample.Profile of Mood States (POMSs, [43]). Used in its Spanish version, adapted and validated by Fuentes et al. [44]. It is a self-report questionnaire to measure mood. The short version was used, with 29 items answered on a Likert-type scale with 5 response options. It includes 5 dimensions: tension (α = 0.83); depression (α = 0.78); anger (α = 0.85); vigor (α = 0.83); fatigue (α = 0.82). The variable Depression EA refers to mood states; this nomenclature is used to distinguish it from other instruments assessing the same variable. In the present study, the reliability coefficients obtained for the dimensions were similar to those reported in the original study by Fuentes et al. [44], suggesting adequate internal consistency for each dimension in our sample.

### 2.3. Procedure

Data collection was carried out first by contacting the triathletes via telephone, during which the purpose and objectives of the study were explained, as well as the confidentiality of their data—both their responses and any previously collected information. Subsequently, they received instructions to complete the questionnaires (Google Forms), and the link was sent via WhatsApp (https://docs.google.com/forms/d/e/1FAIpQLSeyF6x-2ExfLtuZuUvlSwj0-9orL59RV9FhbMJgJEJlGjPdjA/viewform?usp=sf_link (accessed on 21 January 2025)) so that they could fill it out online. In addition, informed consent was obtained, which appeared on the final page of the questionnaire and was accepted by all participants prior to voluntarily taking part in the study.

To ensure quality control in data extraction (i.e., response monitoring), all participants completed every item of the various questionnaires, so no data were excluded. All data collected were entered into an Excel spreadsheet, and, as the instruments used closed-ended scales, no data entry errors were recorded.

The evaluation protocol (variables and instruments) was administered monthly for six months, from February 2023 to July 2023. This period was chosen as it coincided with the start of the competitive season. Each month, data were recorded on injuries sustained, including type and severity, in addition to completing the DASS-21 and POMS questionnaires.

Monthly segmentation of results was mainly intended to establish temporal contingencies between training/competition schedules and the psychological evaluations. This was particularly relevant for the administration of the POMS questionnaire, as it evaluates mood states. More broadly, monthly evaluations across the full test battery helped reduce memory bias, which increases the longer the delay between training/competition events and psychological assessment.

While a repeated-measures protocol was originally planned, not all participants completed assessments each month. Therefore, the study reflects a partially unbalanced panel design, where the same individuals were invited to participate each month, but participation varied. This limited the application of fully longitudinal statistical models and led us to employ cross-sectional comparisons and time-point trend analyses instead. These methodological constraints are acknowledged and discussed in the design and limitations sections.

This study followed the criteria outlined in the Ethics Standards for Research in Sports Sciences [45]. It complies with the ethical and deontological principles of the Declaration of Helsinki [46], as well as all guidelines included in the Code of Good Research Practices of the universities to which the study’s co-authors are affiliated.

### 2.4. Design

In order to partly overcome some of the existing limitations associated with cross-sectional studies, this study employed a descriptive and correlational design with prospective and repeated-measures components. It was conducted as a prospective ex post facto study based on non-probabilistic surveys [47,48].

Although data collection was carried out over six consecutive months (February to July 2023), not all participants completed the assessments at each time point. Therefore, the study reflects a partially unbalanced panel design, which limited the feasibility of applying traditional longitudinal statistical modeling. As a result, the analyses focused on cross-sectional comparisons between injured and non-injured athletes at each time point, as well as identifying temporal trends in psychological variables.

While the original aim was to conduct a fully longitudinal analysis, the variability in monthly participation is acknowledged and has been taken into account in the interpretation of results and in the discussion of the study’s limitations.

### 2.5. Data Analysis

The analyses of this study were carried out using the SPSS 22.0 statistical package. Measures of central tendency (mean) and dispersion (standard deviation and coefficient of variation) were used. In addition, the following analyses were performed: frequency analysis, figure generation (graphs), analysis of differences between two independent samples using the Mann–Whitney U test, analysis of differences among three independent samples using the Kruskal–Wallis test, and correlational analyses using Spearman’s Rho coefficient.

Non-parametric tests were chosen due to the non-normal distribution of several variables, as confirmed through the Shapiro–Wilk test. Effect sizes were calculated for the Mann–Whitney U test using Rosenthal’s r and for the Kruskal–Wallis test using epsilon squared (ε^2^), providing additional information on the magnitude of differences.

At the correlational level, all statistically significant results (*p* < 0.05), as well as those showing a trend toward statistical significance (*p* < 0.10), are reported. As is common in health sciences research (particularly in medicine, psychology, and sports sciences) and in studies of mental health in performance athletes [49,50], it is important to detect significant trends in studies with relatively small sample sizes—such as those using monthly-segmented correlational analyses. This is relevant for two main reasons: to facilitate interpretation of results in a longitudinal study involving multifactorial psychological assessment and to provide valuable guidance for future research that aims to explore these emerging correlational patterns in greater depth.

Only complete cases were included in the analyses; no data imputation was performed, and missing responses were excluded pairwise depending on the statistical test.

## 3. Results

There are 26 athletes registered in February, of which 17 have some type of injury recorded; 9 suffered one injury and 8 two injuries. Considering the first injury recorded, the majority are muscular (*n* = 10; 58.8%), followed by others (*n* = 3; 17.6%), fractures (*n* = 2; 11.8%), and only one case each of tendonitis and contusion (*n* = 1 in both cases, 5.9% respectively). Most of the injuries suffered are mild (*n* = 9; 52.9%) or moderate (*n* = 6; 35.3%), with only one case of serious and very serious in each situation (in both cases, *n* = 1; 5.9%). It can be observed that almost two-thirds of the sample (64.7%) have been affected for an average of at least 18 days, and just over three-fourths (76.5%) for an average of 20.5 days. The mean value of days weighted by injury is 22.91 days, with a very high standard deviation (*SD* = 33.26).

In addition, results have been obtained where the highest scores prorated in the total sample are observed in stress (*M* = 0.90; *SD* = 0.69), followed by depression (*M* = 0.76; *SD* = 0.70), and with the lowest scores in anxiety (*M* = 0.41; *SD* = 0.43). It is important to highlight how athletes reach values of 2.43 in stress, and 2.14 in depression. In turn, the results show the highest scores in vigor (*M* = 2.31; *SD* = 1.03), followed by tension (*M* = 1.71; *SD* = 0.80), fatigue (*M* = 1.48; *SD* = 0.96), and anger (*M* = 1.39; *SD* = 0.90). The lowest scores are obtained in depression (*M* = 1.09; *SD* = 1.02).

The results in Table 1 show how the greater the number of injuries, the more clearly a psychological profile with a greater tendency towards psychopathology (DASS-21) is shown, and a more negative mood state and the opposite of the iceberg profile [51,52]. Significant differences are shown with a *p* < 0.01 in stress and a *p* < 0.05 in depression and in the mood state anger. On the other hand, results with a tendency towards statistical significance (*p* < 0.10) are shown in the fatigue and anxiety mood states.

In Figure 1, it can be observed that athletes with two injuries have a higher differential profile, especially in the anger and tension factor of the POMS and stress and depression of the DASS-21.

At the correlational level, Table 2 shows significant relationships with a *p* < 0.01 between the total number of injuries (TNL) and the factors depression (Rho = 0.533) and stress (Rho = 0.531), finding relationships with a *p* < 0.05 in the anxiety factor (Rho = 0.396). Regarding the total number of injuries (TNL), relationships with a *p* < 0.05 are shown in depression (Rho = 0.473), anxiety (Rho = 0.476), and stress (Rho = 0.439).

Considering the mood states, Table 3 shows significant correlations of *p* < 0.05 between the anger mood state and the TNL and the TGSP. In turn, results with a tendency towards statistical significance (*p* < 0.10) are observed between the depression mood state and the TGSP.

In March 2023, nine athletes completed the tests, and eight of them had at least one injury. Considering the eight injured, the average number of days off work is 16.75 days, with a high standard deviation (*SD* = 25.78). The score is significantly lower than in February. Due to the small number of injured (*n* = 8) and non-injured (*n* = 1) athletes, mean difference analyses and correlational analyses are not performed.

In April 2023, 32 completed the questionnaire, but only 23 were injured. Of these subjects, all reported the severity of the injury, but only 22 responded to injury type 1. In addition, 7 of the 23 athletes reported a second injury. Only considering those who have had one or two injuries, the average number of days affected is 13.65 (*SD* = 17.03), finding values lower than those of February and March. When performing the analysis of differences in means, it can be seen that despite the absence of significant differences when considering the three factors of the DASS-21, it can be seen how the subgroup with two injuries has higher levels of stress compared to the group without injury and with one injury.

Considering the five factors of the POMS questionnaire, in Table 4 it can be observed that there are statistically significant differences in the stress factor (*p* < 0.05), with the highest values being presented by the group with two injuries.

On the other hand, Figure 2 shows a differential profile of athletes with two injuries compared to the other two groups, especially in the stress factor of the DASS-21 and in the mood profile of the POMS (especially in tension and anger). The correlational analysis between the three factors of the DASS-21 and five of the POMS and the two injury indices (TL and IL) shows the absence of significant differences.

Correlational analyses show the absence of significant correlations between total number of injuries (TNL) and total severity of injuries (TGSP) and the three factors of the DASS-21 and the POMS factors.

In May, 26 respondents responded, although only 15 indicated the type of injury at time 1 and the level of severity of injury 1. At the same time, only four responded that they had had a second injury. Considering the analysis of difference in means with the DASS-21, Table 5 shows no significant differences in the three groups. Considering the mood states, the results show significant differences with a *p* < 0.05 in anger and a tendency to statistical significance (*p* < 0.10) in depression. The directionality of the differences is different from that of previous months.

In Figure 3, it can be observed that in the month of May, athletes with one injury have a very high differential peak in the POMS vigor factor, followed by athletes without injuries and very close to these values for athletes with two injuries. In addition, the anger factor has a high score for athletes with injuries compared to the rest (two injuries and no injuries).

Considering the correlational analyses between the two injury indices and the three factors of the DASS-21 questionnaire, no significant correlations or correlations with a tendency toward statistical significance are shown.

Regarding the correlational analyses between the two injury indices and the five factors of the POMS questionnaire, Table 6 shows significant correlations between the number of injuries and anger with a *p* < 0.01 and Rho = −0.478. With a *p* < 0.10, correlations with a tendency toward statistical significance are obtained between the depression factor and the number of injuries (Rho = −0.334).

Considering the month of June, the analysis of the difference in means considering the athletes evaluated without injury, with one injury, and with two injuries did not show significant differences between the groups considering both the DASS-21 and the POMS.

Despite the absence of significant differences, it is clearly observed in Figure 4 how the athletes with two injuries have a more negative and reactive stress and mood profile despite the lack of significant differences.

At the correlational level, no significant correlations are observed between the three DASS-21 factors and the injury indices. However, when considering the five POMS mood states, Table 7 shows correlations of *p* < 0.05 between the total number of injuries (TNL) and the Fatigue factor (Rho = 0.438), and with a tendency to statistical significance (*p* < 0.10) in Tension (Rho = 0.438). Regarding the total severity of injuries (TGSP), symmetrical results are observed to those obtained in TNL: with a *p* < 0.05 there is a difference in Fatigue (Rho = 0.459), and with a *p* < 0.10 in tension (Rho = 0.405).

In July, unlike in the previous months considered, the analysis of the difference in means is carried out using the Mann–Whitney U test for two independent samples (without injury and only one injury) instead of the Kruskal–Wallis test for three samples, since there are no cases of athletes with two injuries.

The analysis of the difference in means considering the three factors of the DASS-21 shows the absence of significant differences. Considering the POMS questionnaire, Table 8 shows differences with a *p* < 0.05 in the Fatigue factor, the magnitude being greater in the non-injured group.

Correlational analyses cannot be performed due to the small sample size.

Table 9 below shows a summary of the significant differences and correlations found based on the two questionnaires and the month considered in the study.

Once the descriptive, mean difference, and correlational analyses for each month have been analyzed, the results obtained with the averages of all the months considered are presented below. The objective is to determine the relationships between the injury rates and the factors considered through the DASS-21 and the POMS.

In order to properly determine the relationships, a correlational analysis is first performed between the two injury indices and the psychological variables of the DASS-21 and POMS, considering the 63 athletes evaluated (both injured and non-injured; that is, athletes with “0” injuries are included). The same analyses are then performed but only including athletes who have had at least one injury.

In Table 10, only one correlation with a tendency towards statistical significance is shown in fatigue (*p* < 0.10; Rho = 0.220).

In order to better determine the relationships between injuries and the averages of the psychological variables, only injured athletes will be selected (*n* = 48).

In Table 11, the results show significant correlations between the average of anxiety and the weighted severity index of the injury (*p* < 0.05; Rho = 0.319); correlations with a tendency to statistical significance (*p* < 0.10) between the number of injuries and the average of anger (Rho = 0.259), and between the weighted severity index of the injury and the average of Depression (Rho = 0.251).

To complement these correlational findings and provide a more integrated overview, Table 12 summarizes the results obtained from the monthly analysis of mean differences and correlations during the six-month follow-up period, as well as the relationships between the average scores of the eight psychological variables and the two injury indices considered. This table facilitates a comprehensive interpretation of the dynamic interactions between psychological factors and injury outcomes across different evaluation points.

## 4. Discussion

The present study aimed to explore the relationship between sports injuries and mental health indicators in triathletes, focusing both on injury frequency and on a weighted injury severity index. The results obtained confirm that injuries significantly impact psychological well-being, as reflected by elevated stress, depression, anger, and fatigue scores, especially in athletes with repeated or severe injuries.

Consistent with previous research [2,4,5], we observed that even moderate injuries are associated with significant psychological symptoms. In February, the month with the highest injury prevalence (65.4% of the sample), statistically significant differences were found in DASS-21 stress (*p* < 0.01), depression (*p* < 0.05), and POMS anger (*p* < 0.05), suggesting that psychological strain peaks when physical impairments are more common. These findings are aligned with those of Ríos-Garit et al. [32], who reported similar increases in anxiety and negative mood among young injured athletes during rehabilitation.

Furthermore, athletes with two injuries consistently showed more reactive psychological profiles across the months, particularly in POMS anger and tension and DASS-21 stress and depression. These patterns reflect the inverted “iceberg profile” described by Morgan [51] and reaffirmed in the sports literature [52], where negative mood states dominate during periods of injury. Parsons-Smith et al. [53] also noted the predictive value of mood profiles for both performance and mental health outcomes in endurance athletes such as triathletes.

The correlation analyses provided robust evidence supporting a psychological burden associated with injury. Across the full sample (*N* = 63), the number of injuries correlated significantly with depression (Rho = 0.533), stress (Rho = 0.531), and anxiety (Rho = 0.396). Among injured athletes only (*n* = 48), the severity of injuries showed significant correlation with anxiety (Rho = 0.319) and tendencies toward significance with depression (Rho = 0.251) and anger (Rho = 0.259). These results are consistent with findings from Gil-Caselles et al. [5] and Olmedilla et al. [34], who emphasized the role of psychological vulnerability—particularly stress and low confidence—in both injury susceptibility and delayed recovery.

Interestingly, the analysis revealed that athletes without injuries sometimes reported more negative mood states (e.g., fatigue, depression) than those with a single injury, suggesting a counterintuitive pattern. This paradox has been addressed in recent studies. Gil-Caselles et al. [5] argue that psychological strain in athletes can arise not only from injuries but also from contextual stressors such as performance expectations, overtraining, or fear of failure. Similarly, Marcouni et al. [54] highlight that mood states like fatigue or tension may fluctuate independently of injury status and should not be used in isolation to assess mental health. These findings underline the complexity of psychological responses in competitive settings and support a multidimensional interpretation of athlete well-being.

Notably, the most frequent injury types observed were muscular injuries (58.8%), and most injuries were classified as mild or moderate. However, even these less severe injuries had a substantial psychological impact. The average time loss was over 20 days in more than 75% of the sample, confirming that functional limitations—even temporary—can contribute to emotional distress [12,24,25].

A particularly relevant finding is the high variability observed in mood states such as Fatigue (*SD* = 0.96) and Tension (*SD* = 0.80), especially during periods when the number of injuries was elevated (e.g., April). The presence of cumulative psychological strain in athletes with two injuries became apparent through higher scores in anger and tension, as seen in Figure 2. These results are supported by prior studies that identified overuse, high training volume, and psychological distress as key contributors to injury vulnerability in triathletes [20,21].

The analysis of monthly data revealed fluctuations in psychological responses. While no significant differences were found in June and July for the DASS-21 scores, the profiles in POMS mood states still revealed more reactive profiles among athletes with two injuries. In May, a notable rise in vigor was observed in athletes with one injury, suggesting possible psychological compensation or increased motivation during recovery, although anger levels remained high. Such fluctuations indicate that positive mood states like vigor may temporarily buffer the effects of injury, but do not eliminate emotional vulnerabilityes, especially if reinjury occurs. This divergent trend in vigor compared to the other POMS subscales may be explained by its positive emotional nature. Unlike tension, fatigue, or anger, which tend to increase under injury-related stress, vigor reflects energy, enthusiasm, and motivation. In some cases, athletes in recovery may experience transient increases in vigor as a form of psychological compensation or renewed engagement with training and rehabilitation. This interpretation aligns with the mood-performance frameworks discussed by Parsons-Smith et al. [53].

These dynamics echo the findings of Reardon et al. [4] and Gouttebarge et al. [2], who emphasize the complex and fluctuating nature of mental health in athletes, influenced by both contextual (competition schedule, training) and personal factors (injury history, emotional coping). Our results support this multidimensional view and provide empirical data showing how both the frequency and severity of injuries are associated with mental health indicators across time.

The use of a weighted injury severity index based on days lost allowed a more nuanced understanding of the injury burden, going beyond binary injury/no-injury distinctions. Although some correlations showed only tendencies toward significance, the consistency of patterns across months and variables strengthens the evidence that injury-related disruption impacts mood and emotional balance.

Moreover, the data support the notion that repeated injuries compound psychological risk. Athletes with two injuries exhibited a greater psychological burden than those with a single injury, even when the individual injuries were classified as mild. This aligns with the observations of Putukian [12], who stressed the cumulative psychological cost of injuries, especially in sports where access to mental health services may be limited or inconsistent.

Finally, our findings strengthen the call made by authors such as Olmedilla and García-Mas [33] for integrated rehabilitation protocols that address not only physical recovery but also emotional readiness. Structured psychological interventions—such as the Psycholight protocol—offer effective frameworks for managing the psychological consequences of sports injuries, improving adherence to rehabilitation programs, and reducing the risk of recurrence.

Although this study focused specifically on triathletes, the psychological patterns observed (particularly the cumulative emotional burden associated with repeated injuries) may be relevant to other endurance sports. Athletes in disciplines such as long-distance running, cycling, or swimming are also exposed to sustained physical demands, repetitive training stress, and similar injury profiles, suggesting that the integration of psychological monitoring into rehabilitation protocols could have broader applicability across endurance contexts.

In light of these results, the inclusion of sport psychology professionals in injury prevention and rehabilitation teams in triathlon is not only advisable but essential. The psychological assessment tools employed in this study proved useful in detecting early signs of emotional distress and in informing personalized support strategies, particularly for athletes with multiple or recurrent injuries.

## 5. Limitations of the Study

This study presents several limitations that should be considered when interpreting the findings. First, although the repeated-measures design allowed for temporal analysis, the monthly participation rate varied, leading to an unbalanced panel. In particular, the relatively small sample sizes in certain months, especially in March and July, may have reduced the statistical power to detect differences and limited the generalizability of the findings for those time points. Future studies are encouraged to conduct a priori power analyses to determine the minimum sample sizes required for detecting meaningful effects over time.

Second, the use of a non-probabilistic sampling method restricts the generalizability of the results beyond the sample analyzed. While this approach is common in exploratory research, it may introduce selection bias.

Third, the reliance on self-report questionnaires for both injury reporting and psychological assessment may be subject to recall bias or social desirability effects, despite efforts to ensure confidentiality and data completeness.

Finally, although the statistical analyses captured significant and emerging trends, the absence of a fully longitudinal dataset precluded the use of advanced modelling techniques to track intraindividual changes over time. Future studies should consider strategies to improve adherence across all time points and increase sample size, particularly in underrepresented subgroups. Additionally, comparative research across different endurance sports (e.g., cycling, long-distance running) could help determine whether the psychological impact of injuries varies by discipline.

## 6. Practical Applications

The findings of this study underscore the importance of integrating psychological assessment into injury monitoring protocols for triathletes. Tools such as the DASS-21 and POMS have demonstrated their utility in detecting early signs of emotional distress that may arise from injury frequency or severity. This information can help coaches, sports psychologists, and medical teams to develop individualized support strategies during both prevention and rehabilitation phases.

Furthermore, recognizing mood fluctuations and stress levels in athletes with multiple or severe injuries enables more accurate return-to-play decisions, potentially reducing the risk of relapse. The use of a weighted injury index (TGSP) also provides a practical model for quantifying injury burden over time, which can be applied in multidisciplinary approaches to athlete care in endurance sports. In this context, coaches should be encouraged to actively monitor emotional well-being post-injury and collaborate with mental health professionals to provide timely support.

## 7. Conclusions

This study identified significant associations between injury incidence and key mental health indicators in triathletes, particularly in relation to stress and depression, with additional correlations observed for anxiety. Athletes who experienced two injuries consistently reported more pronounced emotional disturbances, especially in POMS anger and tension and DASS-21 stress and depression. Although some relationships (such as between injury severity and depression) showed only marginal significance, the overall findings underscore the psychological burden of repeated injuries.

In summary, both injury frequency and severity emerged as meaningful predictors of psychological vulnerability. These results highlight the importance of incorporating mental health screening and support into injury management protocols, reinforcing the need for a holistic and individualized approach to athlete care in triathlon.

## Figures and Tables

**Figure 1 healthcare-13-01657-f001:**
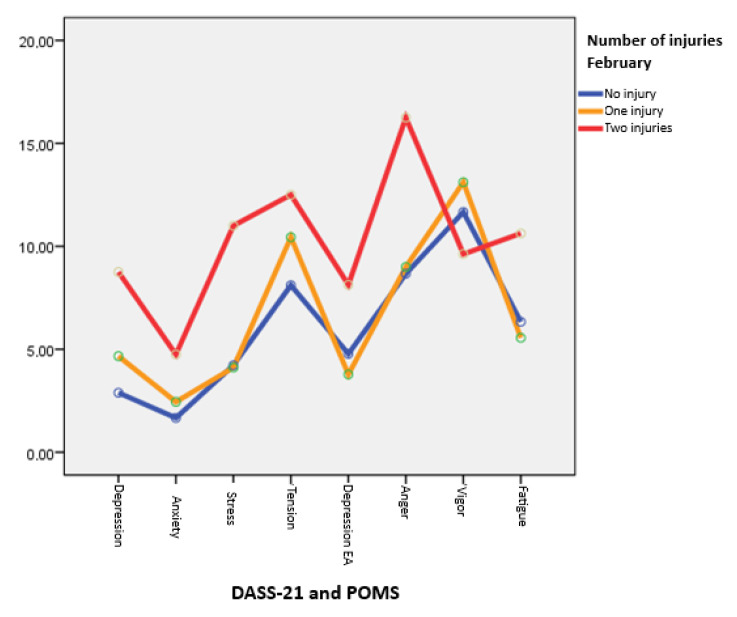
Average DASS-21 and POMS scores for non-injured, one-injury, and two-injury athletes, February 2023. Note. EA Depression Indicator (Mood States, POMS).

**Figure 2 healthcare-13-01657-f002:**
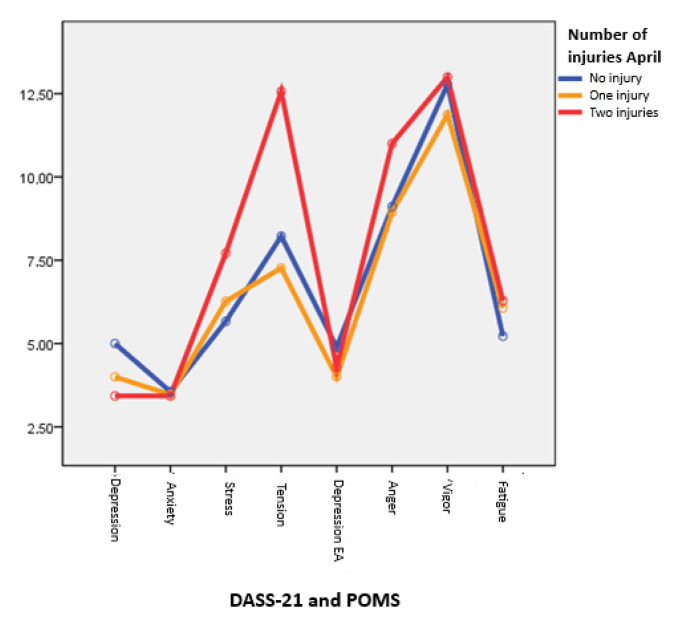
Average DASS-21 and POMS scores for uninjured, single-injured, and double-injured individuals, April 2023.

**Figure 3 healthcare-13-01657-f003:**
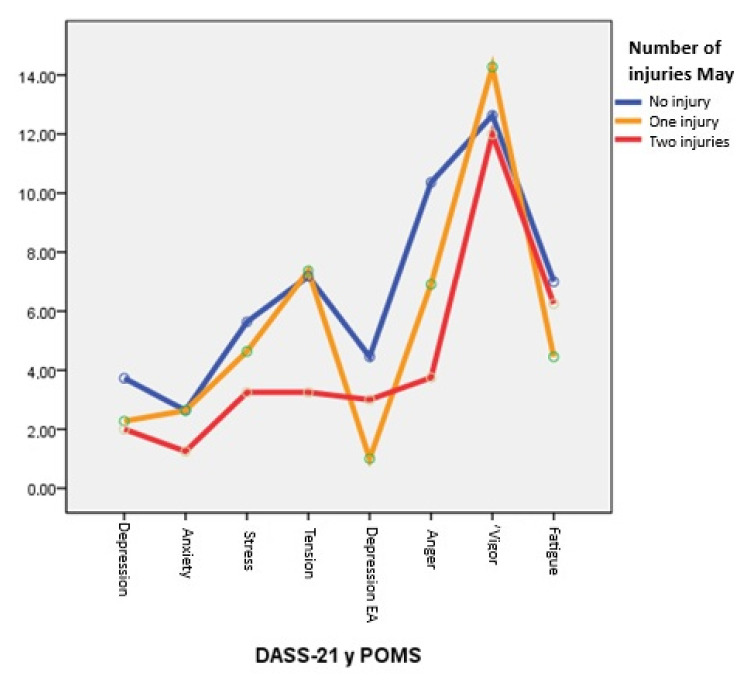
Mean DASS-21 and POMS scores for non-injured, single-injured, and double-injured individuals, May 2023.

**Figure 4 healthcare-13-01657-f004:**
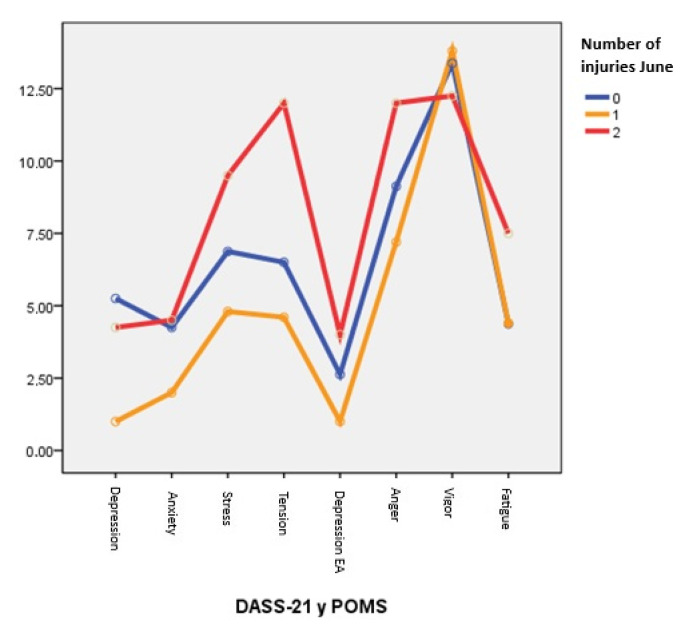
Mean DASS-21 and POMS scores for uninjured, single-injured, and double-injured patients in the month of June.

**Table 1 healthcare-13-01657-t001:** Description and analysis of differences between non-injured, one-injury, and two-injury patients. Month of February (no injury, *n* = 9; one injury, *n* = 9; two injuries, *n* = 8; total, *n* = 26).

		*N*	*M*	*DT*	Average Range	X^2^	*p*
Depression February	No injury	9	2.89	4.20	9.17	7.280	0.026 *
	One injury	9	4.67	4.85	12.89		
	Two injuries	8	8.75	4.17	19.06		
	Total	26	5.31	4.91			
Anxiety February	No injury	9	1.67	1.73	10.83	4.678	0.096 †
	One injury	9	2.44	3.24	11.94		
	Two injuries	8	4.75	3.20	18.25		
	Total	26	2.88	2.98			
Stress February	No injury	9	4.22	4.02	10.44	9.964	0.007 **
	One injury	9	4.11	3.02	10.28		
	Two injuries	8	11.00	4.07	20.56		
	Total	26	6.27	4.80			
Tension February	No injury	9	8.11	5.40	10.39	2.694	0.260
	One injury	9	10.44	4.80	14.06		
	Two injuries	8	12.50	3.16	16.38		
	Total	26	10.27	4.77			
Depression EA February	No injury	9	4.78	4.66	12.50	4.016	0.134
	One injury	9	3.78	4.92	10.67		
	Two injuries	8	8.13	5.41	17.81		
	Total	26	5.46	5.13			
Anger February	No injury	9	8.67	6.61	10.94	7.016	0.030 *
	One injury	9	9.00	6.58	10.78		
	Two injuries	8	16.25	6.34	19.44		
	Total	26	11.12	7.16			
Vigor February	No injury	9	11.67	6.84	14.06	2.367	0.306
	One injury	9	13.111	4.48	15.83		
	Two injuries	8	9.63	3.29	10.25		
	Total	26	11.54	5.15			
Fatigue February	No injury	9	6.33	5.66	12.33	4.918	0.086
	One injury	9	5.56	4.50	10.39		
	Two injuries	8	10.63	2.00	18.31		
	Total	26	7.38	4.78			

† *p* < 0.10; * *p* < 0.05; ** *p* < 0.01.

**Table 2 healthcare-13-01657-t002:** Correlational analysis between total number of injuries (TNL) and total injury severity (TGSP) with the three DASS-21 factors in the month of February (*n* = 26).

		Number Injuries Feb	No. Injuries Feb Weighted
Depression February	Correlation coefficient	0.533 **	0.473 *
	Sig. (bilateral)	0.005	0.015
Anxiety February	Correlation coefficient	0.396 *	0.476 *
	Sig. (bilateral)	0.045	0.014
Stress February	Correlation coefficient	0.531 **	0.439 *
	Sig. (bilateral)	0.005	0.025

* *p* < 0.05; ** *p* < 0.01.

**Table 3 healthcare-13-01657-t003:** Correlational analysis between total number of injuries (TNL) and total severity of injuries (TGSP) with the five POMS factors in the month of February (*n* = 26).

		Number Injuries Feb	No. Injuries Feb Weighted
Tension February	Correlation coefficient	0.326	0.323
	Sig. (bilateral)	0.104	0.107
Depression EA February	Correlation coefficient	0.274	0.348 †
	Sig. (bilateral)	0.175	0.081
Anger February	Correlation coefficient	0.444 *	0.467 *
	Sig. (bilateral)	0.023	0.016
Vigor February	Correlation coefficient	−0.194	−0.203
	Sig. (bilateral)	0.341	0.319
Fatigue February	Correlation coefficient	0.307	0.316
	Sig. (bilateral)	0.127	0.116

† *p* < 0.10; * *p* < 0.05.

**Table 4 healthcare-13-01657-t004:** Descriptives and analysis of differences in means applying the Kruskal–Wallis statistic between non-injured, one injury, and two injuries in the month of April (without injury, *n* = 9; one injury, *n* = 16; two injuries, *n* = 7; total, *n* = 32).

		*N*	*M*	*DT*	Average Range	Chi-Square	Sig.
Tension April	No injury	9	8.22	4.35	16.89	7.276	0.026 *
	One injury	16	6.88	6.00	12.88		
	Two injuries	7	12.57	4.47	24.29		
	Total	32	8.50	5.59			
Depression EA April	No injury	9	4.89	5.28	17.33	0.985	0.611
	One injury	16	3.75	4.78	14.97		
	Two injuries	7	4.29	2.98	18.93		
	Total	32	4.19	4.50			
Anger April	No injury	9	9.11	7.41	15.50	2.366	0.306
	One injury	16	8.56	6.69	14.97		
	Two injuries	7	11.00	4.47	21.29		
	Total	32	9.25	6.38			
Vigor April	No injury	9	12.78	5.04	18.22	0.566	0.753
	One injury	16	12.13	3.81	15.34		
	Two injuries	7	13.00	2.77	16.93		
	Total	32	12.50	3.90			
Fatigue April	No injury	9	5.22	4.87	14.22	1.356	0.508
	One injury	15	6.07	5.71	15.50		
	Two injuries	7	6.29	2.36	19.36		
	Total	31	5.87	4.78			

* *p* < 0.05.

**Table 5 healthcare-13-01657-t005:** Description and analysis of differences between non-injured, one injury, and two injuries in the month of May (no injury, *n* = 11; one injury, *n* = 11; two injuries, *n* = 4; total, *n* = 26).

		*n*	*M*	*DT*	Average Range	Chi-Square	Sig.
Tension May	No injury	11	7.18	6.79	13.50	1.799	0.407
	One injury	11	7.36	4.59	15.09		
	Two injuries	4	3.25	2.36	9.13		
	Total	26	6.65	5.45			
Depression EA May	No injury	11	4.45	4.01	17.05	4.745	0.093 †
	One injury	11	1.00	1.00	10.09		
	Two injuries	4	3.00	3.83	13.13		
	Total	26	2.77	3.35			
Anger May	No injury	11	10.36	7.32	16.86	6.360	0.042 *
	One injury	11	6.91	4.35	12.95		
	Two injuries	4	3.75	0.50	5.75		
	Total	26	7.88	5.91			
Vigor May	No injury	11	12.64	3.29	12.14	0.944	0.624
	One injury	11	14.27	3.13	15.18		
	Two injuries	4	12.00	7.48	12.63		
	Total	26	13.23	3.98			
Fatigue May	No injury	11	7.00	5.16	15.05	1.910	0.385
	One injury	11	4.45	2.46	11.14		
	Two injuries	4	6.25	2.22	15.75		
	Total	26	5.81	3.89			

† *p* < 0.10; * *p* < 0.05.

**Table 6 healthcare-13-01657-t006:** Correlational analysis between total number of injuries (TNL) and total injury severity (TGSP) with the five POMS factors in the month of May (*n* = 26).

		Number Injuries May	No. Injuries May Weighted
Tension May	Correlation coefficient	−0.098	−0.027
	Sig. (bilateral)	0.634	0.894
Depression EA May	Correlation coefficient	−0.334 †	−0.284
	Sig. (bilateral)	0.096	0.160
Anger May	Correlation coefficient	−0.478 *	−0.307
	Sig. (bilateral)	0.014	0.127
Vigor May	Correlation coefficient	0.104	−0.013
	Sig. (bilateral)	0.614	0.948
Fatigue May	Correlation coefficient	−0.090	−0.005
	Sig. (bilateral)	0.661	0.979

† *p* < 0.10; * *p* < 0.05.

**Table 7 healthcare-13-01657-t007:** Correlational analysis between total number of injuries (TNL) and total severity of injuries (TGSP) with the five POMS factors in the month of June (*n* = 20).

		Number of Injuries June	Number of Injuries June Weighted
Tension June	Correlation coefficient	0.438 †	0.405 †
	Sig. (bilateral)	0.061	0.085
Depression EA June	Correlation coefficient	0.208	0.163
	Sig. (bilateral)	0.379	0.493
Anger June	Correlation coefficient	0.150	0.257
	Sig. (bilateral)	0.539	0.287
Vigor June	Correlation coefficient	−0.286	−0.114
	Sig. (bilateral)	0.222	0.633
Fatigue June	Correlation coefficient	0.463 *	0.459 *
	Sig. (bilateral)	0.040	0.042

† *p* < 0.10; * *p* < 0.05.

**Table 8 healthcare-13-01657-t008:** Descriptives and analysis of differences between non-injured, one injury, and two injuries in July (no injury, *n* = 4; one injury, *n* = 4; total, *n* = 8).

	Total Injuries July	*n*	*M*	*DT*	Average Range	Sum of Ranks	U de Mann–Whitney	Sig.
Tension July	No injury	4	8.00	6.88	3.63	14.50	4.500	0.309
	One injury	4	12.75	4.03	5.38	21.50		
Depression EA July	No injury	4	3.00	2.58	4.25	17.00	7.000	0.766
	One injury	4	3.50	4.04	4.75	19.00		
Angel July	No injury	4	11.00	8.29	4.88	19.50	6.500	0.655
	One injury	4	7.00	3.56	4.13	16.50		
Vigor July	No injury	4	2.75	0.85	5.50	22.00	4.000	0.248
	One injury	4	1.70	1.29	3.50	14.00		
Fatigue July	No injury	4	6.50	3.70	6.50	26.00	0.000	0.019 *
	One injury	4	1.50	0.58	2.50	10.00		

* *p* < 0.05.

**Table 9 healthcare-13-01657-t009:** Summary table of the results obtained in the analysis of difference in means and correlational analysis in the six months of evaluation with the two injury indices.

	Difference of Means		Correlations	
	DASS-21	POMS	DASS-21	POMS
February	Depression (*) Stress (**) Anxiety (†)	Anger (*) Fatigue (†)	Depression-PT (**) e IP (*) Anxiety-TL (*)-IP(*) Stress-TL(**) IP (*)	Anger-LT (*)-IT (*) Depression-IT (†) ***** fatigue is eliminated
March	*n* = 8 sample insufficiency	*n* = 8 sample insufficiency	*n* = 8 sample insufficiency	*n* = 8 sample insufficiency
April	No significant differences are shown	Tension (*)	No significant differences are shown	No significant differences are shown
May	No significant differences are shown	Anger (*) Depression (†)	No significant differences are shown	Anger-LT (*) Depression-LT (†)
June	No significant differences are shown	No significant differences are shown	No significant differences are shown	Tension-LT (†)-IT (†) Fatigue-LT (*) y IT (*)
July	No significant differences are shown	Fatigue (*)	*n* = 8 sample insufficiency	*n* = 8 sample insufficiency

† *p* < 0.10; * *p* < 0.05; ** *p* < 0.01; *** *p* < 0.001.

**Table 10 healthcare-13-01657-t010:** Correlational analysis applying Spearman’s Rho between the injury rates (TL), the total average severity index, and the average values of the eight psychological variables considered (*n* = 63; athletes without injury and with at least one injury).

		Total Injuries	Total Weighted Injuries
Depression mean	Correlation coefficient	0.034	0.036
	Sig. (bilateral)	0.795	0.784
Anxiety mean	Correlation coefficient	0.032	0.129
	Sig. (bilateral)	0.804	0.316
Stress mean	Correlation coefficient	0.132	0.120
	Sig. (bilateral)	0.307	0.353
Tension mean	Correlation coefficient	0.144	0.131
	Sig. (bilateral)	0.259	0.305
Depression EA mean	Correlation coefficient	0.016	0.060
	Sig. (bilateral)	0.900	0.651
Anger mean	Correlation coefficient	0.177	0.133
	Sig. (bilateral)	0.166	0.298
Vigor mean	Correlation coefficient	−0.149	−0.173
	Sig. (bilateral)	0.243	0.174
Fatigue mean	Correlation coefficient	0.184	0.220 †
	Sig. (bilateral)	0.151	0.086

† *p* < 0.10.

**Table 11 healthcare-13-01657-t011:** Correlational analysis applying Spearman’s Rho between the injury rates and the average values of the eight psychological variables considered.

		Total Injuries	Total Weighted Injuries
Depression mean	Correlation coefficient	0.146	0.148
	Sig. (bilateral)	0.321	0.314
Anxiety mean	Correlation coefficient	0.162	0.319 *
	Sig. (bilateral)	0.271	0.027
Stress mean	Correlation coefficient	0.216	0.187
	Sig. (bilateral)	0.140	0.202
Tension mean	Correlation coefficient	0.216	0.187
	Sig. (bilateral)	0.140	0.202
Depression EA mean	Correlation coefficient	0.180	0.251 †
	Sig. (bilateral)	0.238	0.096
Anger mean	Correlation coefficient	0.259 †	0.183
	Sig. (bilateral)	0.076	0.214
Vigor mean	Correlation coefficient	−0.128	−0.183
	Sig. (bilateral)	0.384	0.213
Fatigue mean	Correlation coefficient	0.156	0.222
	Sig. (bilateral)	0.294	0.133

† *p* < 0.10; * *p* < 0.05.

**Table 12 healthcare-13-01657-t012:** Summary table of the results obtained in the analysis of mean differences and correlational analysis in the six months of evaluation and the average scores of the eight psychological variables with the two injury indices.

	Difference of Means		Correlations	
	DASS-21	POMS	DASS-21	POMS
February	Depression (*) Stress (**) Anxiety (†)	Anger (*) Fatigue (†)	Depression-PT (**) e IP (*) Anxiety-TL (*)-IP(*) Stress-TL(**) IP (*)	Anger-LT (*)-IT (*) Depression-IT (†) ***** fatigue is eliminated
March	*n* = 8 sample insufficiency	*n* = 8 sample insufficiency	*n* = 8 sample insufficiency	*n* = 8 sample insufficiency
April	No significant differences are shown	Tension (*)	No significant differences are shown	No significant differences are shown
May	No significant differences are shown	Anger (*) Depression (†)	No significant differences are shown	Anger-LT (*) Depression-LT (†)
June	No significant differences are shown	No significant differences are shown	No significant differences are shown	Tension-LT (†)-IT (†) Fatigue-LT (*) y IT (*)
July	No significant differences are shown	Fatigue (*)	*n* = 8 sample insufficiency	*n* = 8 sample insufficiency
Average over 6 months	Not applicable	Not applicable	With *n* = 63 No significant differences are shown With *n* = 48 Anxiety-IT (*)	With *n* = 63 Fatigue-IT (†) With *n* = 48 Depression-IT (*) Anger-TI (†)

† *p* < 0.10; * *p* < 0.05; ** *p* < 0.01; *** *p* < 0.001.

## Data Availability

The raw data supporting the conclusions of this article will be made available by the authors, without undue reservation.

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
