# Peer review of "Impact of Injury Frequency and Severity on Mental Health Indicators in Triathletes: A Repeated-Measures Study"

_healthcare, 2025, doi:10.3390/healthcare13141657_

Round 1

Reviewer 1 Report

Comments and Suggestions for Authors

Dear authors, thank you very much for the effort and work you have put into this manuscript. Without a doubt, the topic is highly relevant and contributes to the growing body of scientific knowledge regarding injuries and their psychological impact on athletes—particularly in triathletes.

However, I have several comments and suggestions that, in my opinion, must be addressed in order to enhance the quality and impact of the paper. I have listed them below and also provided an annotated PDF as a supporting tool for the suggested corrections.

  • Introduction: The introduction is well-structured and well-documented. It provides a clear rationale for the study and effectively highlights its importance.

  • Methods Section: There are several issues that require clarification:

    • The study reports a sample of 63 triathletes, yet the total number of subject data points across the six months appears to be 121 (if counted correctly). There is no explanation of how repeated measures were handled or how many athletes were evaluated in each month, which makes it difficult to interpret the longitudinal nature of the results.

    • The study design is misplaced in the manuscript. Furthermore, the design is described as “descriptive, correlational, and longitudinal.” However, the analyses performed are limited to group comparisons and cross-sectional correlations, with no true longitudinal analysis applied.

    • While the sample size may be small for more complex regression models, the authors could consider analyzing data across all time points using repeated measures statistics or small worthwhile change approaches to detect meaningful changes over time.

  • The results section is currently difficult to follow due to its complex structure.

  • There is an excessive number of tables and figures, many of which do not provide additional insight beyond what is already stated in the text. I recommend consolidating and reducing the number of tables/figures and grouping the data more effectively.

  • Conducting more in-depth statistical analyses and reporting them clearly in the results section could significantly increase the scientific value and impact of the manuscript.

  • Interpretation of the results needs to be more careful and precise. While the findings from February and April align with expectations regarding the impact of injury on mood and mental health, from May onwards the results show a reversed trend—non-injured athletes appear to report more negative mood states. This requires a thorough explanation in the discussion.

  • It is important not to conflate the mental health indicators assessed by the DASS-21 with mood states assessed by the POMS. A clearer separation of these constructs in the interpretation would enhance the clarity of the findings. This distinction may also help explain the unexpected mood results in the latter months—possibly due to accumulated fatigue, training load, or other stressors.

  • The discussion is limited in scope and lacks a strong connection between the study’s findings and previous literature.

  • Avoid overly brief paragraphs that do not add substantive interpretation.

  • Provide a more detailed explanation of the study's results, including potential reasons behind the observed trends.

  • Interpret marginal or statistically non-significant results cautiously, and clearly communicate these limitations to readers.

  • The conclusion should adopt a more reflective and reserved tone, acknowledging the limitations of the study while emphasizing its relevance and potential to guide future research.

Please consider these suggestions to improve the clarity, rigor, and scientific contribution of your manuscript. I hope they are helpful in preparing a stronger and more impactful version of your work.

Kind regards.

Reviewer 2 Report

Comments and Suggestions for Authors

Thank you very much for allowing me to review this manuscript. I congratulate the authors for their work, it is a very interesting topic and has great applicability. However, some issues need to be resolved before final publication.

Abstract

Well written. In addition to the mean, I would include the standard deviations. Also the p-values.

Introduction

Very good sharing of relevant aspects and scientific evidence regarding injuries and mental variables. At the end of the introduction I see the aims, but I would also like to see the hypotheses.

Methods

Is the sample representative? Please include the sample size calculation.

In the injury questionnaire, the average days associated with each injury severity also came from the previous study [40]? It would be nice to know if this methodology has been used before or is used by the authors for the first time

“2.1 Design”. I believe this is an error and it should be 2.4.

Results

They are well formulated, although somewhat lengthy due to the division by each month in which data collection took place. Nevertheless, they are accurate.

However, an important question arises regarding the data analysis performed. The analysis appears to be entirely based on establishing correlations and comparing injured versus non-injured athletes. Yet, if I have correctly understood the study design, the same athletes were measured across different months (although not all athletes were assessed every time, the majority were). Was the possibility of conducting a repeated measures analysis considered to examine how the athletes evolved over time, depending on whether they sustained injuries during the given periods? I raise this point because focusing solely on whether an athlete was injured or not may overlook other confounding variables that could be influencing the results, and the observed effects might not be due solely to the injury itself.

Discussion

The section is well developed in line with the stated objectives and the results achieved. If a hypothesis was presented in the introduction, it should also be addressed in the discussion.

It is recommended not to include statistical values (e.g., means, standard deviations, p-values) in this section.

Include a section on practical applications. The results provide substantial information—take advantage of this to offer something useful and applicable for the readers.

Thank you once again for allowing me to review this excellent work. I congratulate the authors and hope they are able to consider my comments and suggestions.

Reviewer 3 Report

Comments and Suggestions for Authors

Dear Editor,

Please find attached my review of the manuscript titled "Impact of injury frequency and severity on mental health indicators of triathletes: a longitudinal study." The study addresses a relevant topic in sports psychology, though improvements in design reporting, statistical transparency, and generalizability are recommended. I hope the comments will support the authors in refining their manuscript.

PAGE 1

 Line 2–3 – Add a clearer and more precise title that specifies the study design.

Line 13–14 – The objective should include a specific hypothesis or whether the study is exploratory.

Line 15–19 – Methods should mention the study design explicitly.

PAGE 2

Line 29–31 – Clearly define mental health and provide more epidemiological context. Include specific data on the prevalence of mental health issues among endurance athletes.

Line 35–37 – State the rationale for focusing on triathletes early in the introduction.

PAGE 4

Line 103–104 – Add a flowchart showing the number of participants assessed, excluded, and analyzed.

Line 110–115 – Report reasons for exclusion and define missing data strategy.

PAGE 5

Line 195–198 – The study design should be stated more clearly, particularly the observational nature.

PAGE 6

Line 201–217 – Statistical methods are adequately described but need more details on: Handling of missing data; effect size reporting; justification for using non-parametric tests (e.g., Mann-Whitney).

PAGE 8

Line 218–236 – Use visual aids (e.g., stacked bar graphs) to show injury and stress level progression.

PAGE 13

Line 373–385 – Clarify the limitation due to small sample size in July.

PAGE 15

Line 393–417 –  Provide 95% CIs for correlation coefficients where possible.

PAGE 18

Line 423–426 – Restate the design as observational, not just “descriptive.”

Line 495–497 – Statement on anger could be supported by reference to specific POMS values. Include table or figure references to direct the reader to relevant data.

PAGE 19

Line 518–534 – Indicate how these can be generalized to other endurance sports.

PAGE 20

Line 536–538 – Better articulate the sample size limitation across months, and suggest statistical power analysis for future research.

Reviewer 4 Report

Comments and Suggestions for Authors

Following this report, we evaluated the scientific qualifications, strengths, and points for improvement of the article “Impact of injury frequency and severity on mental health indicators of triathletes: A longitudinal study”. This study examined the effects of injury frequency and severity on mental health indicators among triathletes. A detailed evaluation of the article is presented below, and the sections are discussed separately.

The hypotheses of this study can be stated clearly. For example, a statement such as “As the frequency and severity of injuries increase, triathletes' levels of depression, anxiety and stress are predicted to increase” could be added.

More arguments could be presented about why triathlon constitutes a special sample compared to other sports.

More information can be provided on the response rates and missing data management of the online surveys used in the data collection process.

Due to the study's longitudinal design, dropout rates and their possible impact on the results can be discussed.

Explain why nonparametric tests (Mann-Whitney U Kruskal and–Wallis) were preferred for statistical analysis.

Small sample sizes for some months (for example, March and July) may limit statistical power. The impact of this on the generalizability of the results should be discussed.

Why “Vigor”, one of the POMS sub-dimensions, shows a different trend than the other dimensions can be analyzed more deeply.

Practical applications of the results could be emphasized more. For example, it could be suggested that coaches should provide mental health support after injury.

Suggestions for future studies can be made more specific. For example, comparative studies in different sports branches can be suggested.

Round 2

Reviewer 1 Report

Comments and Suggestions for Authors

Dear authors of the manuscript titled "Impact of Injury Frequency and Severity on Mental Health Indicators of Triathletes: A Repeated-Measures Study",

First, I would like to thank you for taking into account the suggestions and observations made during the first round of review. I have carefully read both the revised manuscript and your responses to the comments, including those areas where you felt changes were not feasible. I find your explanations regarding the statistical treatment to be reasonable, and I agree that addressing these limitations explicitly in the manuscript is an appropriate way to acknowledge the potential impact on your findings.

Upon this second review, I can confirm that the manuscript has improved in clarity and the discussion section has been notably strengthened. However, there are still some concerns regarding the results and their interpretation. While I have included specific comments directly in the PDF, I would also like to summarize them here:

  • Some tables do not present information for athletes with two injuries.

  • Specifically, the tables for June and July lack data from the subgroup of athletes with two injuries. Additionally, non-injured athletes appear to report worse mood states compared to those with one injury, not only in the vigor dimension but also in fatigue.

  • Table 5 presents a formatting issue: a partial image of a p-value (including a "<" symbol and an asterisk) appears on the left side of the table.

  • The DASS-21 results show a significant difference only for athletes with two injuries in February. In subsequent months, no significant differences were found between injured and non-injured athletes. On the other hand, the POMS dimensions revealed greater variability. While mood states assessed by POMS may relate to mental health, we must be cautious not to conflate the two constructs directly. For example, the depression dimension of POMS showed only a marginally significant difference in May, with higher scores among non-injured athletes. This suggests a complex and somewhat counterintuitive pattern, where non-injured athletes reported worse mood states than those with injuries.

This does not diminish the relevance of injury as a factor in athletes' mental health, but rather underscores the importance of considering other psychological variables—such as coping skills, performance outcomes throughout the season, and external pressures—that may interact with or mediate the relationship between injury status and mental state.

Lastly, I recommend relocating the conclusion section to follow the limitations and practical applications, in line with conventional scientific writing structure.

I hope you find this comments as favorable to a last improvement of the document.

Kind regards

Reviewer 4 Report

Comments and Suggestions for Authors

Dear authors

Thanks for your effors.

Best regards.

Author Response

Thank you very much for your work. Best regards